# Genome-wide meta-analysis associates *HLA-DQA1/DRB1* and *LPA* and lifestyle factors with human longevity

Peter K. Joshi et al.[#]

Genomic analysis of longevity offers the potential to illuminate the biology of human aging. Here, using genome-wide association meta-analysis of 606,059 parents' survival, we discover two regions associated with longevity (*HLA-DQA1/DRB1* and *LPA*). We also validate previous suggestions that *APOE*, *CHRNA3/5*, *CDKN2A/B*, *SH2B3* and *FOXO3A* influence longevity. Next we show that giving up smoking, educational attainment, openness to new experience and high-density lipoprotein (HDL) cholesterol levels are most positively genetically correlated with lifespan while susceptibility to coronary artery disease (CAD), cigarettes smoked per day, lung cancer, insulin resistance and body fat are most negatively correlated. We suggest that the effect of education on lifespan is principally mediated through smoking while the effect of obesity appears to act via CAD. Using instrumental variables, we suggest that an increase of one body mass index unit reduces lifespan by 7 months while 1 year of education adds 11 months to expected lifespan.

#A full list of authors and their affliations appears at the end of the paper

Longevity is of interest to us all, and philosophers have long speculated on the extent to which it is pre-determined by fate. Here we focus on a narrower question—the extent and nature of its genetic basis and how this inter-relates with that of health and disease traits. In what follows, we shall use longevity as an umbrella term. We shall also more specifically refer to lifespan (the duration of life) and long-livedness (living to extreme old age, usually defined by a threshold, such as 90 years). Up to 25% of the variability in human lifespan has been estimated to be genetic[1], but genetic variation at only three loci (near *APOE*, *FOXO3A* and *CHRNA3/5*)[2–5] have so far been demonstrated to be robustly associated with lifespan.

Prospective genomic studies of lifespan have been hampered by the fact that subject participation is often only recent, allowing insufficient follow-up time for a well-powered analysis of participant survival. On the other hand, case-control studies of long-livedness have had success[2, 3, 6] and some technical appeal (focussing on the truly remarkable), but such studies can be limited and costly in their recruitment. We recently showed that the extension of the kin-cohort method[7] to parental lifespans, beyond age 40, of genotyped subjects could be used to detect genetic associations with lifespan with some power in genomically British participants in UK Biobank (UKB)[4]. Here we extend that approach in a genome-wide association meta-analysis (GWAMA) to discovery across UKB European- and African-ancestry populations and 24 further population studies (LifeGen), mainly from Europe, Australia and North America, to search for further genetic variants influencing longevity. We then use those GWAMA results to measure genetic correlations and carry out Mendelian randomisation (MR) between other traits and lifespan seeking to elucidate the underlying effects of disease and socio-economic traits on longevity, in a framework less hampered by confounding and reverse causality than observational epidemiology.

## Results

**Genome-wide association study**. In total, 606,059 parental lifespans were available for analysis, of which 334,974 were already complete (Table 1).

In our GWAS of 586,626 European parental lifespans, we find four regions *HLA-DQA1/DRB1*, *LPA*, *CHRNA3/5* and *APOE*, in which the lead SNPs rs34831921, rs55730499, rs8042849 and rs429358, respectively, associate with survival at genome-wide significance ($p < 5 \times 10^{-8}$) (Table 2, Fig. 1a, b, Fig. 2a–d). The two previously unreported loci, rs34831921 (*HLA-DQA1/DRB1*) and rs55730499 (*LPA*), both showed statistically significant, directionally consistent, evidence of association at the proxy SNPs in strongest LD in the largest (5406 cases, 15,112 controls) publicly available set of GWAS summary statistics for extreme long-livedness (CHARGE-EU 90+)[6], with $p < 0.0035$ for both

SNPs. As our GWAS results were of the observed effect of offspring genotype on parent phenotype and the actual effect of carrying an allele for the individual concerned (rather than their parent) is twice that observed in a parent-offspring kin-cohort study[4], all reported effect sizes (and their standard errors) throughout this manuscript have been doubled to give the estimated effect size in the allele carriers themselves. The hazard ratios for one copy of the minor alleles were 0.942 and 1.074 for rs34831921 (*HLA-DQA1/DRB1*) and rs5573049 (*LPA*), respectively, corresponding to an increase/decrease in lifespan of ~ 0.6/0.7 years for a carrier of one additional copy of the minor allele.

We meta-analysed our results with the CHARGE-EU 90+ longevity GWAMA[6] summary statistics using $Z$-scores and equal weights for each study, reflecting their similar statistical power. We found strengthened signals, substantially at *APOE* (rs4420638, $p = 5.4 \times 10^{-41}$) and slightly in the *LPA* region (rs1045587, $p = 2.05 \times 10^{-11}$). No improvement of statistical significance was observed in the *HLA-DQA1/DRB1* region, where there were no SNPs in strong LD with the lead LifeGen SNP, nor was there an increase in significance near *CHRNA3/5*. However, in this meta-analysis one further region near *AKAP7/ EPB41L2* on chromosome 6 just reached genome-wide significance (rs1919453, A allele frequency = 0.36, $p = 4.34 \times 10^{-8}$; Fig. 1c, Supplementary Fig. 1), and the observed hazard ratio (SE) for the minor allele was 0.976 (0.0056) in LifeGen alone.

In our study of 9359 father and 10,074 mother lifespans in participants with African ancestry, no SNPs were genome-wide (GW) significant in the analysis of both parents combined. However, we found one GW significant signal (rs10198124, G allele frequency 0.39 in African subjects), in an intergenic region of chromosome 2 associating with lifespan for fathers (HR (SE) for G allele = 1.22 (0.0354), $p = 1.66 \times 10^{-8}$), with a consistent direction of association in all 9 cohorts studied. No association was observed at this SNP in African mothers, or fathers and mothers of European ancestry (HR (SE) = 0.97 (0.038), 1.01 (0.007) and 1.00 (0.008), $p = 0.51$, 0.21 and 0.77, respectively (Fig. 1d, Supplementary Fig. 2A–D).

**Cross-validation of candidate genes**. We next attempted to validate 13 candidate genes identified in previous longevity studies. In our study, only three of these genes showed statistically significant, directionally consistent evidence ($p < 0.0003$, two-sided test) of association; *CDKN2A/B*, *SH2B3* and *FOXO3A* (Fig. 3, Supplementary Fig. 3 and Supplementary Data 3). For *SH2B3* and *FOXO3A* our estimated effect sizes are concordant with those reported from the most robust (i.e., narrowest 95% confidence interval (CI)) previous study. However, for *CDKN2A/B*, the 95% CI for our estimate is entirely below that from the more robust of the two studies considered.

**Table 1 Summary of the LifeGen parental lifespans**

| Ancestry | Parent | Count | | | Mean age | | |
|---|---|---|---|---|---|---|---|
| | | Alive | Dead | Total | Alive | Dead | All |
| African | Father | 2435 | 6924 | 9359 | 72.4 | 70.4 | 70.9 |
| African | Mother | 4185 | 5889 | 10,074 | 73.1 | 70.7 | 71.7 |
| European | Father | 113,611 | 178,017 | 291,628 | 62.9 | 71.2 | 68 |
| European | Mother | 150,854 | 144,144 | 294,998 | 66.2 | 75.1 | 70.5 |
| | ALL | 271,085 | 334,974 | 606,059 | | | |

Summary statistics for the 606,059 parental lifespans that passed phenotypic QC (in particular, parent age > 40) and were analysed here. In practice, fewer lives than these were analysed for some SNPs, as a SNP may not have passed QC in all cohorts (in particular within cohort MAF > 1%). The mean age of alive parents across European cohorts was reduced by the large iPSYCH cohort, of relatively younger subjects and thus parents, who were predominantly alive (mean father/mother age among the alive parents in iPSYCH was 52.4/50.4)

**Table 2 Four regions associated with lifespan at genome-wide significance and replication via proxy SNPs in CHARGE**

| rsid | Gene | a1 | Freq a1 | N(000) parent | HR a1 | SE | P-value | Years | Proxy | r² | CHARGE P | Dir. |
|------|------|----|---------|---------------|-------|-----|---------|-------|-------|----|---------|------|
| rs34831921 | HLA-DQA1 /DRB1 | A | 0.09 | 481 | 0.942 | 0.011 | 4.18 E-08 | 0.6 | rs3129720 | 0.39 | 0.003 | + |
| rs55730499 | LPA | T | 0.083 | 563 | 1.074 | 0.011 | 8.67 E-11 | −0.7 | rs10455872 | 0.97 | 0.002 | − |
| rs8042849 | CHRNA3/5 | C | 0.356 | 567 | 1.046 | 0.006 | 3.75 E-14 | −0.4 | rs9788721 | 0.98 | 0.951 | − |
| rs429358 | APOE | C | 0.142 | 556 | 1.091 | 0.008 | 1.44 E-27 | −0.9 | rs6857 | 0.69 | 2E-20 | − |

a1 the effect allele, CHARGE, CHARGE European GWAS for survivorship beyond age 90 vs. younger controls,[6] CHARGE P, the p-value for the two-sided test of association between proxy and long-livedness in CHARGE, Dir. direction of effect of a1 in CHARGE: " + " means long-livedness increasing, "−" means long-livedness decreasing, Freq. frequency, N(000) count (thousands of parents with lifespan and subject genotype information), HR, Hazard Ratio, P p-value for the Wald test of association between imputed dosage for a1 and lifespan, Proxy, the closest proxy SNP in CHARGE, r² the linkage disequilibrium between the discovery SNP and its CHARGE proxy, in the 1000 genomes EU panel, SE, Standard Error, Years the number of additional years of lifespan expected for a carrier of one additional copy of a1. There are four overlapping cohorts between the two studies; EGCUT, NTR, PROSPER and RS1, but only RS1 contributed cases to the CHARGE: out of all 5406 cases analysed in CHARGE, 892 cases (from RS1) overlapped the 300,000 genotyped subjects studied in discovery and the phenotyped individuals were in any case not the same

No statistically significant (p > 0.22, two-sided test) evidence of association was found for the other 10 genes. In all cases (with the possible exceptions of ABO and 5q33) our estimates of the odds ratio were close to 1 and our 95% CI did not include previous estimates, suggesting, at least for the remaining 8 SNPs (at or near CAMK4, C3orf21, GRIK2, IL6, RGS7, CADM2, MINPP1 and ANKRD20A9P), that our non-replication did not arise solely from lack of power.

Consistent with our previous reports[4], we found age-specific and sex-specific effects of the lead SNPs in the APOE and CHRNA3/5 loci. For APOE, the hazard ratio (SE) of the lead SNP was 1.07 (.01) for men and 1.13 (.01) for women, whereas for CHRNA3/5 it was 1.07 (.01) for men and 1.04 (.01) for women (Fig. 4a). Conversely, for APOE, hazard ratios stratified by age were 1.06 (.01) for ages 40−75 and 1.14 (.01) for ages 75+, whereas for CHRNA3/5 they were 1.08 (.01) for 40−75 and 1.03 (.01) for age 75+ (Fig. 4b), with similar patterns when stratifying by age and sex at the same time, (Fig. 4c), although the distinctions between men and women for CHRNA3/5 disappeared beyond age 75. For LPA, CDKN2B and SH2B3, there was no statistically significant evidence of age-specific or sex-specific effects, while the HLA and FOXO3 variants showed age but not sex-specific effects (Fig. 4a, b), with the HLA locus having a greater effect at younger ages (40−75) while, conversely, the FOXO3 locus had greater effect at older ages(75+).

We tested the four SNPs identified in the discovery phase (Table 2) for association with other ageing traits, using PhenoScanner[8], an on-line tool which searches 88 complex trait GWAMAs and three GWAS catalogues. For the SNP in the LPA region, associations were found with blood lipids and coronary traits. For the SNP in the HLA region, we found associations with rheumatoid arthritis and Crohn's disease. For the CHRNA3/5 region, we found associations with traits which associate with smoking behaviour: nicotine dependence, lung cancer, chronic obstructive pulmonary disease and schizophrenia. Finally, for the APOE region, we saw associations with Alzheimer's disease, age-related macular degeneration, blood lipids, adiposity, cardiac and cognitive ageing traits (Supplementary Data 4).

**Genetic correlation of complex traits with lifespan.** We estimated the genetic correlation between 113 complex quantitative and disease susceptibility traits and lifespan using LD Score regression[9]: 46 showed meaningful genetic correlations (rg) with lifespan (statistically significant, |rg| > 0.15). The most strongly correlated with mortality were coronary artery disease (CAD) and cigarettes smoked per day, rg (SE) = 0.66 (0.05) and 0.58 (0.11), respectively. Those most negatively correlated were years of schooling and former vs. current smoker, rg (SE) = −0.47 (0.05) and −0.64 (0.09), respectively (Supplementary Fig. 4, Supplementary Data 5). Lung cancer, type 2 diabetes and insulin resistance also correlated relatively strongly with earlier mortality, while increased age at first birth, openness to experience (a personality trait reflecting curiosity vs. caution, determined by questionnaire) and high-density lipoproteins (HDL) cholesterol were correlated with later death.

Estimates for rg between 9 traits and mortality and their 95% CI fell wholly within the range [−0.15, 0.15], which we have labelled not meaningfully correlated with lifespan. These were femoral neck and lumbar spine bone mineral density, serum creatinine, extreme height, height, bipolar disorder, schizophrenia, autism spectrum disorder and platelet count. For the remaining 55 traits, there was insufficient statistical power to distinguish whether the rg fell within or outside [−0.15, 0.15].

Given the similarity in definition of many traits (e.g., obesity classes) and the strong correlations between others, we clustered the 46 traits which showed a significant and meaningful rg into nine clusters. Positive genetic correlations with mortality for the clusters ranged from 0.68 (smoking) to 0.17 (rheumatoid arthritis and breast cancer), whilst negative correlations varied from −0.50 (education) to −0.15 (age at menarche); (Fig. 5, Supplementary Data 5). We found that the beneficial trait clusters for education and happiness group together, as do a core group of factors (obesity, dyslipidemia/waist-hip ratio (DL/WHR), type 2 diabetes, CAD and smoking) which show stronger correlation not only to mortality but also among each other, while albuminuria and blood pressure seem to form their own risk cluster. We next considered whether and to what extent the observed correlations between mortality and the trait clusters are mediated through other clusters, using partial correlations. In most cases, there was relatively little difference between correlations and partial correlations with mortality (Supplementary Table 1) and the direction of effects remained the same. On the whole, the correlation of each risk cluster is therefore not mainly mediated via other clusters. However, the entire correlation of the DL/WHR cluster with lifespan was 0.41, whereas its partial correlation was −0.18, implying that one or more of the other clusters influenced the genetic correlation, likely CAD with which it is strongly correlated and whose partial correlation did not fall in the same manner. Similarly, the entire correlation of the education cluster with lifespan fell from −0.50 to −0.18 as a partial correlation, in this case apparently due to mediation through smoking behaviour. Blood pressure and age at menarche also showed reductions in partial rg, to near zero for age at menarche, consistent with mediation by other traits.

**Causal relationships with lifespan.** Finally, we used MRbase[10] and further summary statistics for breast cancer (BCAC[11]) and C-reactive protein (CHARGE-CRP[12]) made available to us to

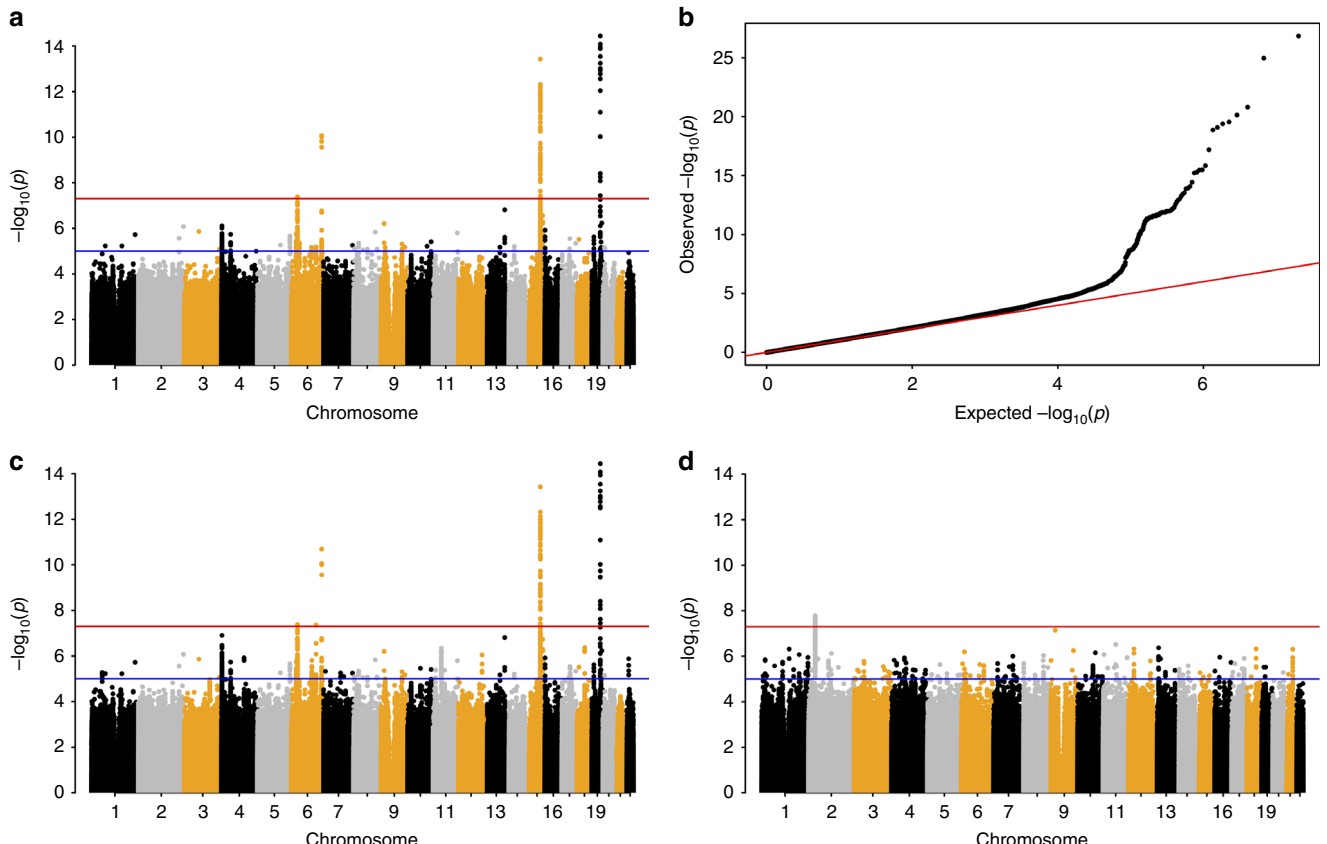

**Fig. 1** Genome-wide associations with parental lifespan. Association analysis was carried out using imputed allelic dosages. **a** Manhattan plot for LifeGen European ancestry, with both parents combined; **b** Q–Q plot comparing the expected (under the null hypothesis) and actual (observed) –log10 p-values for results in **a**; **c** Manhattan plot of meta-analysis of LifeGen Europeans (both parents combined) with CHARGE-EU 90+ published summary statistics[6]. The meta-analysis used Z-scores and equal weights, as suggested by the near equality (9.5/9.4, LifeGen, CHARGE) of Z-test statistics at rs4420638. The additional (just) GW significant SNP lies between the two chromosome 6 hits in **a**; **d** Manhattan plot for LifeGen African fathers only. In Manhattan plots, the y-axis has been restricted to 15 to aid legibility

perform two-sample Mendelian randomisation to investigate causal influences on lifespan. Of more than 90 tested phenotypes, seven risk factors (cigarettes smoked per day, HDL cholesterol, LDL cholesterol, fasting insulin, systolic blood pressure and CRP) and six disease susceptibilities (Alzheimer's disease, breast cancer, CAD, ischaemic stroke, squamous cell lung cancer and type 2 diabetes) significantly associated with mortality (Table 3). Smoking causally reduced lifespan by 6.8 years for lifelong smoking of one pack of 20 cigarettes a day, BMI reduced life by 7 months per unit, while education causally increased lifespan by 11 months for each further year spent studying. In contrast to the genetic correlations (rg CRP: mortality = 0.35), genetically raised CRP seems to have a life-lengthening effect: 5.5 months of increased lifespan per log mg/L.

We compared the relative strengths of these different phenotypic effects on lifespan using a measure independent of scale: extrapolating the genetic effects across the interquartile phenotypic range. Variation in smoking and systolic blood pressure had the strongest causal life-shortening effects (5.3 and 5.2 years, respectively), followed by fasting insulin, body mass index and CAD, while years of education showed by far the most beneficial effect (4.7 years), when comparing the estimated effect of moving from the first to the third quartile of the phenotype distribution. Similarly, we estimate moving from the bottom to the top of the interquartile phenotypic range of CRP increases lifespan by 0.7 years.

## Discussion
We replicated previous findings of genome-wide significant associations between longevity and variants at CHRNA3/5 and APOE and discovered two further associations, at LPA and HLA-DQA1/DRB1, with replication of the further associations in a long-livedness study. We found no evidence of our lead SNPs at the CHRNA3/5, LPA and HLA-DQA1/DRB1 loci associating with traits other than smoking behaviour, cardio-metabolism and rheumatoid arthritis, respectively, while finding more pleiotropy at APOE. We also robustly replicated previous work suggesting associations with longevity at CDKN2A/B, SH2B3/ATXN2 and FOXO3A. We found no evidence of association between lifespan and the other 10 loci previously found to suggestively associate with lifespan, despite apparent power to do so. We showed strong negative genetic correlation between CAD, smoking and type 2 diabetes and lifespan, while education and openness to experience were positively genetically correlated. Using MR, we found that moving from the 25th to 75th percentile of cigarettes per day, systolic blood pressure, fasting insulin and BMI causally reduced lifespan by 5.3, 5.2, 4.1 and 3.8 years, respectively, and similarly moving from the 25th to 75th percentile of educational attainment causally extended lifespan by 4.7 years. Strikingly, we also found that increased CRP increases lifespan, as a causal effect, the reverse of its correlation.

Lipoprotein(a) is a spherical lipoprotein carrying cholesterol and triglycerides in the bloodstream[13]. Variation in LPA has

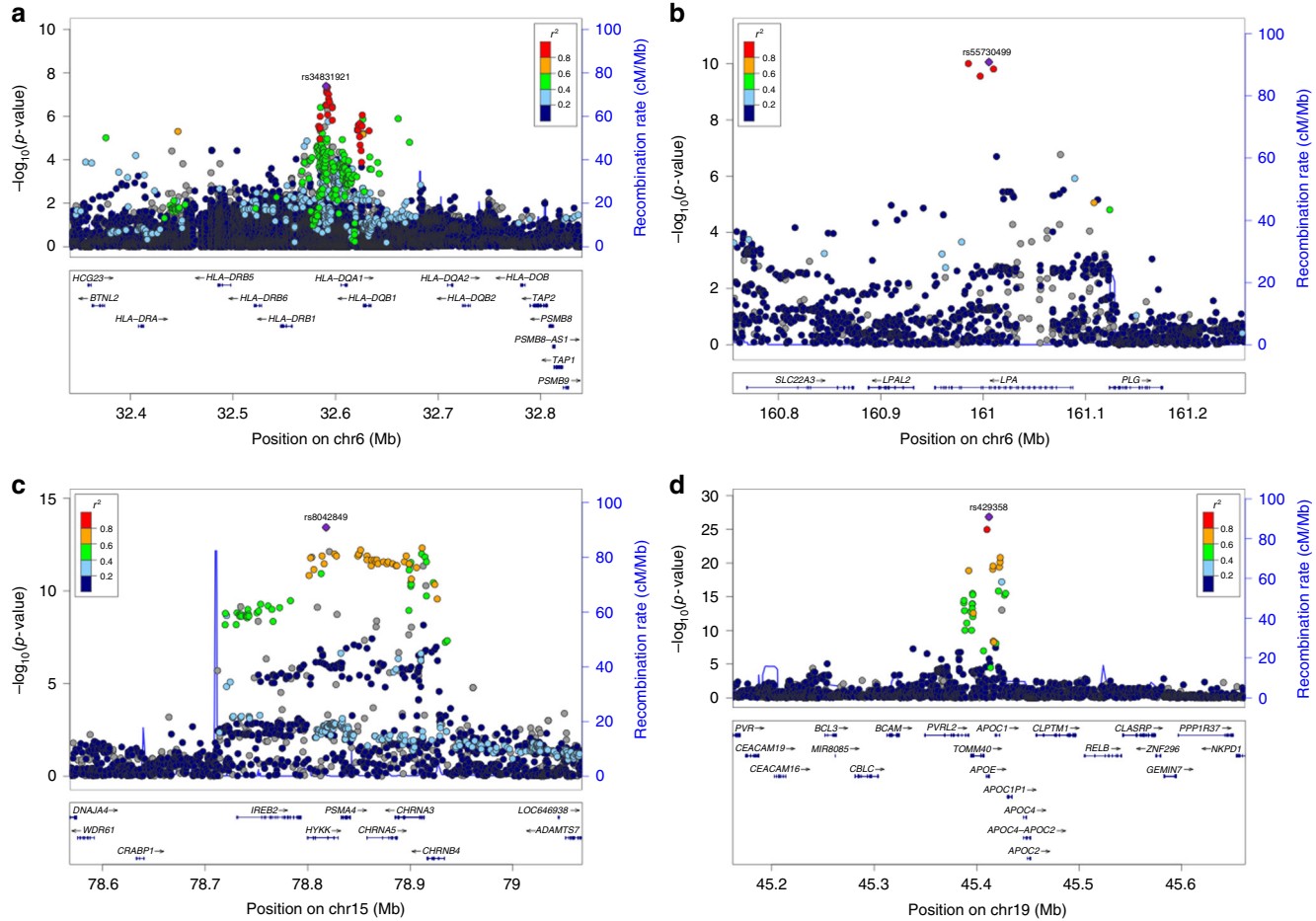

**Fig. 2** Locus zoom plots for four genome-wide significant associations with lifespan. Results from the meta-analysis of subjects of European ancestry analysis, for both parents combined. The displayed p-value corresponds to that of a two-sided test of association between the SNP and parent lifespan under the Cox model. **a** The rs34831921 variant, at the HLA-DQA1/DRB1 locus, P = 4.18E-08. **b** The rs55730499 variant, at the LPA locus, P = 8.67E-11. **c** The rs8042849 variant, at the CHRNA3/5 locus, P = 3.75E-14. **d** The rs429358 variant, at the APOE locus, P = 1.44E-27

been extensively studied[14], and found to influence cardiovascular disease[15] and type 2 diabetes[16]. A close proxy to our lead SNP (rs10455872, $r^2 = 0.97$) has been strongly associated with decreased Lp(a) size and increased Lp(a) plasma concentration and is one of the strongest predictors of coronary heart disease risk with an odds ratio of 1.7 per allele, consistent across populations[17], all suggesting that rs55730499 affects mortality by increasing Lp(a) levels and susceptibility to cardiovascular events.

The large major histocompatibility complex (MHC) encompasses *HLA-DQA1/DRB1*. MHC class II genes encode components of the antigen-presenting apparatus and are the most polymorphic region of the human genome. Genes within the MHC have previously been associated with many autoimmune conditions and other traits, including psoriasis[18], rheumatoid arthritis[19], multiple sclerosis[20] and T1D[21]. In a recent informed GWAS of longevity, Fortney et al.[22] identified, but failed to replicate, two variants close to the *HLA-DRA* locus[22].

The *FOXO3A* locus has been repeatedly reported by other studies[3, 23] as associating with extreme longevity. Variant rs3800231, which exhibits the strongest association in our data, seems to exert its beneficial effect on people aged above 75 but may have a neutral, or deleterious effect at younger ages, supporting the consensus that *FOXO3A* plays a putative role in extreme longevity and general health into old age. This contrasts our findings for the *CHRNA3/5*, *LPA*, *HLA-DQA1/DRB1* loci,

where effects appear to be specific to disease susceptibility, rather than general ageing. The *CDKN2A/B* locus at 9p21 has previously been associated with CAD[24], while the missense allele rs3184504-T we identified within the *SH2B3/ATXN2* locus has been previously associated with increased risk for type 1 diabetes[25], diastolic blood pressure[26] and several autoimmune conditions[27–29].

The failure to replicate previous findings for lifespan increases at *ABO* and 5q33.3/*EBF1* may be due to a combination of limited power in our study, despite its size, and a degree of winner's curse in previous findings. However, for *CAMK4*, *C3orf21*, *GRIK2*, *IL6*, *RGS7*, *CADM2*, *MINPP1* and *ANKRD20A9P*, our findings appear inconsistent with the previous work, suggesting those findings were either false positive associations, or differences in effects are due to the differences between the types of lives studied by us and other studies.

The use of different cohorts from a diverse range of countries with common shared ancestry is common in GWAMA and potentially gives rise to heterogeneity in effect sizes, whatever the trait under consideration. However, a study of lifespan is perhaps particularly susceptible to such effects, as mean lifespans vary by cohort (Supplementary Data 2) and genetic effects might vary by environment. Nonetheless, such heterogeneity is not relevant under the null hypothesis (effect size = 0 in all cohorts) and so will not have induced false positives. On the other hand,

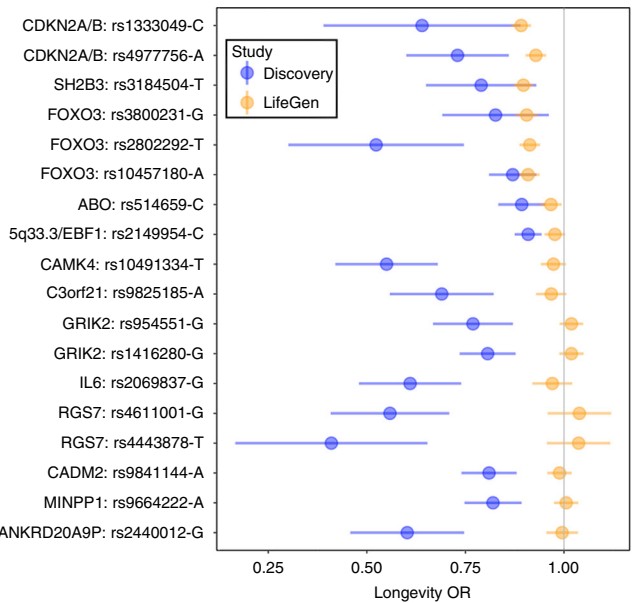

**Fig. 3** Validation of associations reported elsewhere by lookup in LifeGen. A search of recent literature suggested the gene regions shown here were most likely to harbour associations with lifespan, beyond the four loci identified in Table 2, which are further explored in the Discussion. The most powerful LifeGen analysis (i.e., European ancestry, father and mother combined) was used for validation. The odds ratio (OR) for extreme long-livedness is presented for the reported life-shortening allele (i.e., the OR for long-livedness < 1) in the original study, but not necessarily in LifeGen. The LifeGen OR of being long-lived was estimated empirically on the assumption that the relationship between the LifeGen observed hazard ratio (HR) and the OR is stable across allelic effects, with APOE results from LifeGen and CHARGE-EU 90+ 6 being used to estimate the ratio of ln HR to ln OR (−4.7). These estimates will only fully align with the published ORs if the shape of the effect on lifespan is similar to APOE, as is true under the proportional hazards assumption, nonetheless the pattern is suggestive. Further details are shown in Supplementary Data 3

heterogeneity may have reduced power and estimated effect sizes should perhaps be considered as (sample-weighted) averages over the cohorts participating.

The lack of observed genetic correlation between mortality and schizophrenia is perhaps surprising, given the known increased risk of early death due to schizophrenia[30], however, here we study lifespan after the age of 40, where the effect of schizophrenia relative to other causes of mortality is less pronounced. We conjecture that a study of early mortality might show a different pattern, but believe the parent-offspring kin-cohort method would be less suitable, as parents would have to survive beyond reproduction to be available for study. The albuminuria cluster, which correlated with mortality, is understood to be a consequence of poor glomerular filtration arising from chronic kidney disease, often attributable to diabetes or high blood pressure[31]. Our finding that the happiness cluster (depressive symptoms and subjective well-being) has a beneficial correlation with lifespan (rg = 0.24), is in line with a recent meta-analysis which has shown a life-lengthening effect of subjective well-being on lifespan[32]. Similarly, depression has been shown to increase mortality, and is one of the strongest quality-adjusted life expectancy losses, twice as much as better-studied risk factors such as smoking, heart disease, stroke and diabetes[33]. Our results thus reinforce the importance of public policy focusing not only on physical health but also on general well-being in order to increase life expectancy and quality[34].

In general the results of the MR analyses appear consistent with those of the LD score regression estimates. This might be expected since the main difference is that MR compares two phenotypes using just a small number of SNPs which the underlying GWAS were powered to find, and LD score regression uses the whole genome. Nevertheless, as a result the latter may indicate a shared heritable confounding factor, rather than a causal effect, which appears to be the case for our CRP results, as the measured effect of CRP on lifespan is in the opposite direction to the genetic correlation. CRP's effects per se are not well understood, but our results lead us to speculate it may have a protective function, rising in the presence of disease, rather than causing it, despite observational associations with disease and consequent attempts to develop a drug to reduce it[35]. If true, this pattern is somewhat analogous to findings for the N-terminal fragment of pro-BNP, which is a protective molecule, but observationally positively associates with cardiac failure and adverse cardiovascular outcomes[36]. Our finding that a reduction in one BMI unit leads to a 7-month extension of life expectancy, appears broadly consistent with those recently published by the Global BMI Mortality Collaboration, where great effort was made to exclude confounding and reverse causality[37]. We also found each year longer spent in education translates into approximately a year longer lifespan. When compared using the interquartile distance, risk factors generally exhibited stronger effects on mortality than disease susceptibility. Although both CAD and cigarette smoking show a very similar genetic correlation with lifespan, the measured effect of smoking is twice as large as that of CAD, perhaps because smoking influences mortality through multiple pathways.

Our results show that longevity is partly determined by the predisposition to common diseases and, to an even greater extent, by modifiable risk factors. The genetic architecture of lifespan appears complex and diverse and there appears to be no single genetic elixir of long life.

## Methods

**Genome-wide association.** As is conventional in GWAMA, analysis was carried out locally at each cohort and then meta-analysed centrally. Initial phenotype and genotype quality control were carried out in accordance with local standards, with variants imputed to 1000 Genomes (typically phase 1, version 3). Cohort characteristics, including genotyping and imputation methods and summary statistics of the parental lives analysed are described in Supplementary Datas 1 and 2. Study protocols were approved by the relevant committees for each of the local cohorts. Written informed consent was obtained from each participant in each study.

We conducted an association test between parental survival (age and alive/dead status) and offspring genotype. To do so, survival traits were transformed into residuals, permitting analysis as quantitative traits. To facilitate standardisation across the GWAS consortium, residuals for GWAS were calculated in accordance with the analysis plan set out below using a common R protocol distributed to all groups. These residual traits were then tested for association in a GWAS over the imputed SNP panel.

Parents who died below the age of 40 were excluded. Analysis was thus of survivorship beyond the age of 40. Association testing was conducted under the following Cox Proportional Hazards Model[38],

$$h(x) = h_0(x)e^{\beta X + \gamma_1 Z_1 + \ldots + \gamma_k Z_k}$$

$h_0$ is the baseline, $\beta$ the hazard $\log_e$ ratio associated with $X$ (the effect allele count) and $Z_1, \ldots, Z_k$ the other variables fitted i.e., subject sex, and the first 10 PCs of genetic structure along with each studies' usual further covariates, such as batch or assessment centre.

Rather than fit the full model in one step, we calculated Martingale residuals of the Cox model (excluding $X$). Martingale residuals[39] are,

$$\widehat{M}_i = \delta_i - \widehat{\Lambda}_0(\tau_i)e^{\widehat{\gamma}_1 Z_1 + \ldots + \widehat{\gamma}_k Z_k}$$

where $\delta_i$ and $\tau_i$ are the parent status (1—dead/ 0—alive at assessment date) and age of the $i$th individual, $\widehat{\gamma}_1 \ldots \widehat{\gamma}_k$ are effect estimates of, $Z_1, \ldots, Z_k$. Where the allele count, $X$, has an effect, $\widehat{M}_i$ has a linear association with it[39].

However, although these residuals are associated proportionately with the hazard ratio and thus permit statistical hypothesis testing, their relationship with

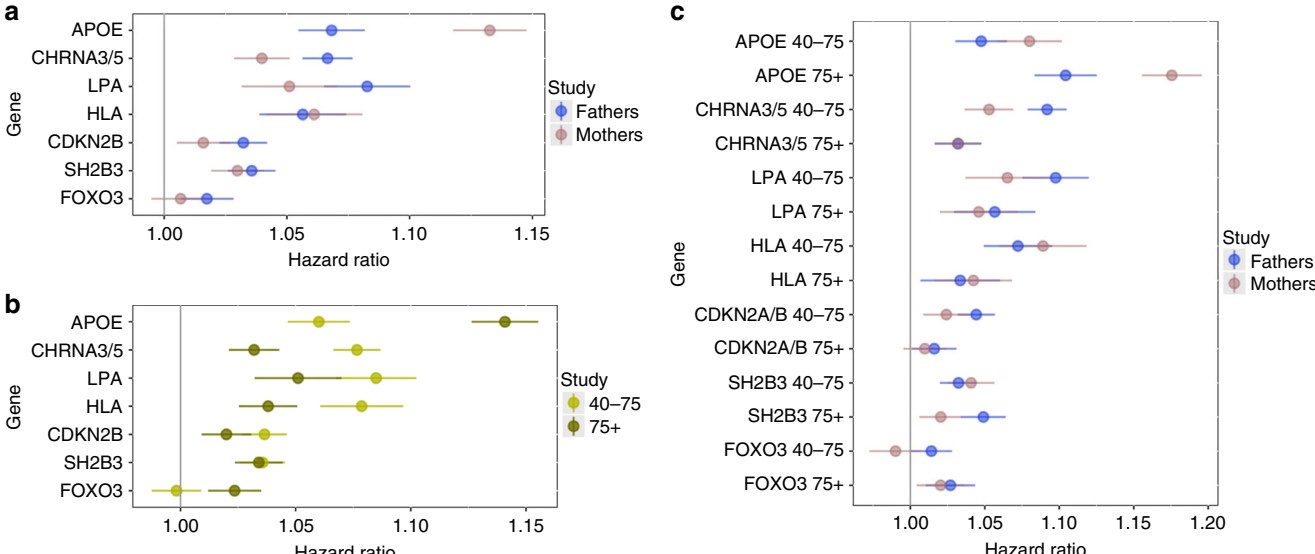

**Fig. 4** Age-specific and sex-specific effects of the 4 GWS associations in LifeGen and the validated candidate loci. The four GWS and three suggestive replicated loci were analysed for age-specific and sex-specific effects on lifespan. **a** The variants at APOE and CHRNA3/5 exhibit sexually dimorphic effects on parental mortality, while all other variants exhibit more modest often non-significant sex-specific differences. **b** The effects of each gene on male and female lifespan were meta-analysed and studied in the cases that died aged between 40 and 75 or after 75. APOE exerts a much greater effect in the older age group, while most of the other genes exhibit the opposite effect. FOXO3 appears neutral, if not positive, in the earlier age group. **c** Effects on mortality were studied in both age groups for both sexes. APOE has the strongest effect on females aged 75+, CHRNA3/5 acts on males aged 40–75 and all other genes display more ambiguous trends

the hazard ratio depends of the (parent) population structure, in particular the proportion dead. The Martingale residuals were therefore scaled up by 1/(proportion dead) separately for each parent gender, to give a residual trait with a 1:1 correspondence with the hazard ratio[39]. This transformed trait was then tested for association with each SNP separately under the following (additive) model,

$$P = \beta X + e$$

where $\beta$ is the effect size of the SNP (and an estimate of the HR) and $X$ the non-reference allele count of the marker, with $e$ being normally distributed and independent. Despite this efficient approach, runtimes in UK Biobank were still potentially onerous, so RegScan 0.2[40] was used there as it is ideally suited for multiple, residualised traits in large data sets.

For cohorts with significant relatedness, all but one subject amongst relatives with coefficient of kinship > 5% were excluded, to create a (smaller) unrelated population, in preference to conventional (potentially more powerful) mixed modelling among family based studies. This was done because the genomic relationship matrix among offspring does not precisely reflect the genetic covariance among parental traits. As an example, consider the offspring of two brothers: the correlation between genetic values of the father trait is 0.5, but of the mother trait is 0, while the Genetic Relationship Matrix (GRM) entry would be 0.25 in both cases. The GRM thus does not fully express covariance among (parent) trait values across subjects, so the smaller unrelated population was used. Exceptions to this were for CILENTO, ERF, GeneSTAR, MICROS and OGP, where it was impractical to exclude relatives, and mixed modelling was used.

After preparing GWAS results locally, cohorts submitted these to the central team for meta-analysis. The central meta-analysis was carried out in METAL, with QC following that of Easy-QC[41], but sometimes more conservative, as follows. UK Biobank data were read into METAL[42] first, standardising all subsequent input alleles to that imputation. SNPs with mismatching alleles in other GWAS were rejected. SNPs were removed from a cohort's GWAS if the minor allele frequency for that cohort was < 0.01. As all studies had in excess of 500 lives, this meant that minor allele count exceeded 10 Alleles with an info score (observed variance in dosage/expected under HWE) < 0.3 were excluded. Each GWAS was checked for systematic errors in allele coding/frequencies and test statistics for SNPs passing QC. After QC, SNP counts were 13,689,868 for European fathers, 13,643,373 for European mothers, 20,305,364, for African fathers and 20,296,065 for African mothers.

African and European ancestries were meta-analysed separately, as were the results for each parental sex, using inverse variance meta-analysis in METAL. Double genomic control was applied. The median λ for 78 GWAS was 0.998 and the maximum was 1.048, suggesting good control for stratification. The highest λ was for UK Biobank—genomically British, the most powered study. After the first level of genomic control, results were meta-analysed by inverse variance, while keeping continental ancestry separate and parental sex separate. The λ applied was

1.034, 1.023, 1.027, 1.028, for European fathers, mothers, African fathers, mothers, respectively. Finally, within continent across parent inverse variance meta-analysis was applied. As expected, due to environmental correlation among spouses, there was some inflation: λ of 1.107 and 1.094, for Europeans and Africans, respectively, giving two final combined meta-analyses (African and European) for both parents combined, subject to double genomic control.

These GWAS results were of the observed effect of offspring genotype on parent phenotype. The actual effect of carrying an allele for the individual concerned (rather than their parent) is twice that observed in a parent-offspring kin-cohort study[4]. All reported effect sizes throughout this manuscript were therefore doubled to give the estimated effect size in the allele carriers themselves. The effect of hazard ratios on lifespan was calculated from survival curves of the Cox model by each cohort. The weighted average effect of hazard ratio on lifespan across all cohorts and both sexes was that a 1% reduction in hazard extended expected lifespan by 0.108 years. To avoid an undue sense of precision, and in accordance with an actuarial rule of thumb, where applicable, hazard ratios were converted to estimated effects on lifespan using a 10% HR: −1 year of lifespan ratio.

Genome-wide significant European lead SNPs at each QTL were then looked up in the largest independent GWAS of lifespan with published summary statistics, for survivorship beyond age 90 vs. younger controls (CHARGE-EU 90+)[6]. None of the lead SNPs were present in that dataset, so proxy SNPs in strongest LD were chosen using LDlink[43], with European populations selected. The SNP showing the highest $r^2$ with each LifeGen lead SNP was extracted from the CHARGE GWAS. The Rotterdam study was part of both GWAMAs, but the trait measured was in different people. In our study, we considered the lifespan of parents, whereas the long-livedness analysis was in the offspring. The LifeGen and CHARGE-EU 90+ GWAMAs were then meta-analysed using p-values and direction of effect (after reversing the sign of effect for CHARGE to convert longevity to mortality) with equal weights placed on each GWAMA, using METAL. The choice of equal weights was made, rather than weights reflecting sample size, because (i) the CHARGE extreme case-control approach is more powerful per sample than parent lifespan Cox modelling, and comparison of $n$ is not straightforward, (ii) the Z-test statistics for rs4420638 (the most significant SNP overlapping in both studies) were similar: 9.4 and 9.5 for CHARGE and LifeGen, respectively, for the same $n$, indicating similar overall power.

We used PhenoScanner[8] to search for other trait associations with our lead SNPs. Pheno Scanner settings were: rsid, Catalogue = GWAS, p-value cutoff = .001, proxies = 1000 G, $r^2 = 0.6$.

A review of recent literature was conducted for SNPs that have been associated with longevity and lifespan by other researchers, to see if we could validate their results. Nine papers published since 2008 were selected: Broer et al.[3], Deelen et al.[6], Emanuele et al.[44], Flachsbart et al.[45], Fortney et al.[22], Malovini et al.[46], Newman et al.[47], Willcox et al.[23] and Zeng et al.[48]. Variants with MAF above 1% in 1000 Genomes, from these papers were taken forward, if they exhibited genome-wide significance ($p < 5 \times 10^{-8}$) or they had suggestive associations that were replicated.

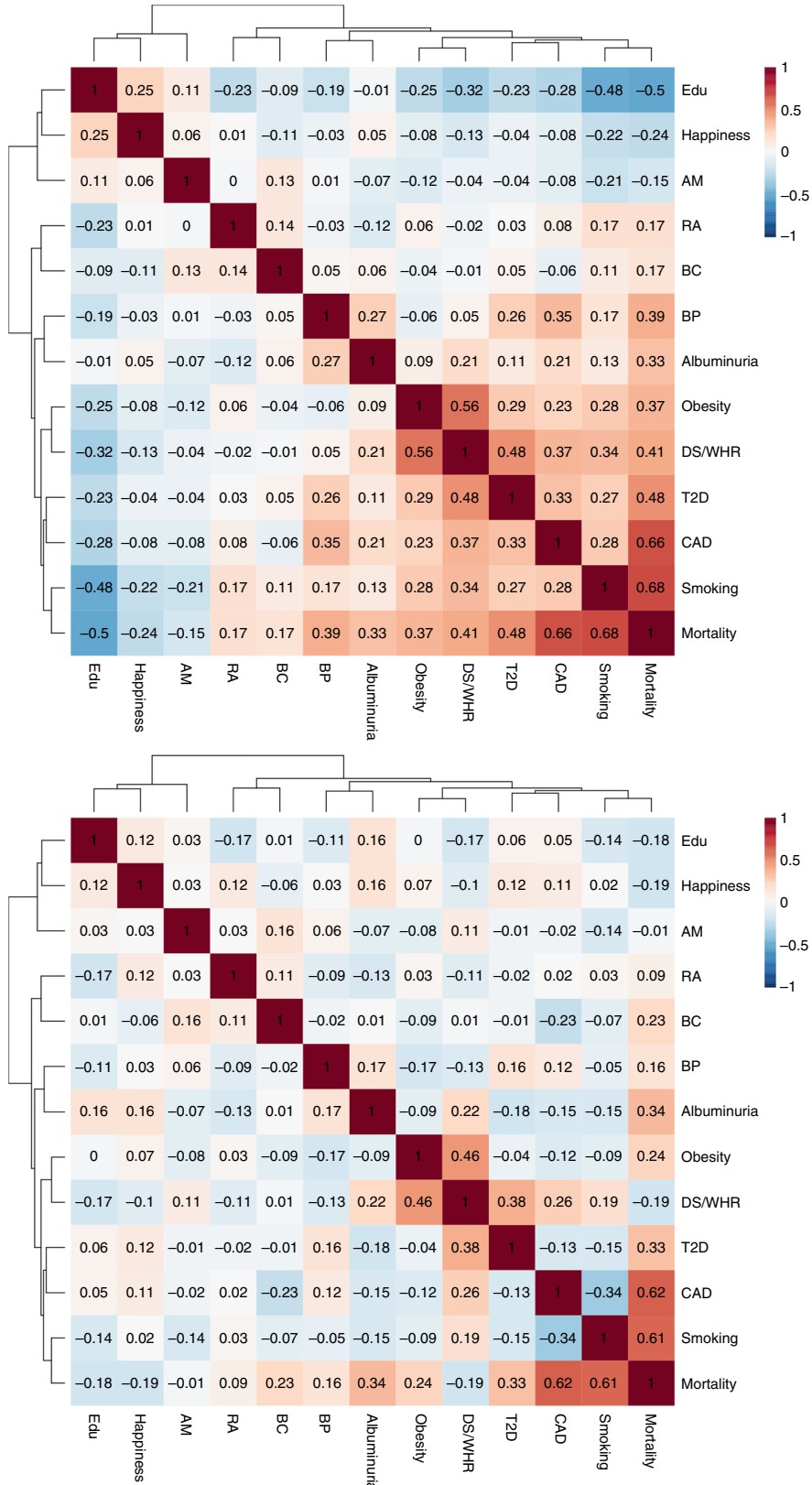

**Fig. 5** Genetic correlations between trait clusters that associate with mortality. The upper panel shows whole genetic correlations, the lower panel, partial correlations. T2D, type 2 diabetes; BP, blood pressure; BC, breast cancer; CAD, coronary artery disease; Edu, educational attainment; RA, rheumatoid arthritis; AM, age at menarche; DL/WHR Dyslipidaemia/Waist-Hip ratio; BP, blood pressure

**Table 3 Mendelian randomisation associations for the 19 traits with lifespan**

| Exposure | SNPs in the IV | Beta | SE | P-value | Egger pleiotropy P | SD | Years per exposure unit | Interquartile effect in years |
|---|---|---|---|---|---|---|---|---|
| *Risk factor* | | | | | | | | |
| Body mass index SD (kg/m$^2$) | 65 | 0.279 | 0.04 | $2.26 \times 10^{-12}$ | 0.4 | 4.77 | 0.584 | 3.8 |
| Years of schooling SD (years) | 64 | −0.348 | 0.054 | $9.42 \times 10^{-11}$ | 0.039 | 3.71 | −0.937 | −4.7 |
| Cigarettes smoked per day | rs12914385 | 0.034 | 0.005 | $6.47 \times 10^{-10}$ | – | 11.7 | 0.338 | 5.3 |
| HDL cholesterol SD (mg/dL) | 39 | −0.106 | 0.044 | 0.017 | 0.793 | 15.5 | −0.068 | −1.4 |
| LDL cholesterol SD (mg/dL) | 17 | 0.101 | 0.042 | 0.017 | 0.82 | 38.7 | 0.026 | 1.4 |
| Fasting insulin log pmol/L | 6 | 0.389 | 0.176 | 0.027 | 0.823 | 0.79 | 3.89 | 4.1 |
| SBP mmHg | rs381815 | 0.02 | 0.009 | 0.031 | – | 18.9 | 0.204 | 5.2 |
| CRP log mg/L | 39 | −0.046 | 0.021 | 0.033 | 0.073 | 1.08 | −0.458 | −0.66 |
| DBP mmHg | 3 | 0.029 | 0.015 | 0.056 | 0.248 | | | |
| Omega-3 fatty acids (SD) | rs145717049 | −0.229 | 0.182 | 0.208 | – | | | |
| Total cholesterol SD (mg/dL) | 11 | 0.036 | 0.068 | 0.597 | 0.348 | | | |
| Triglycerides SD (mg/dL) | 18 | 0.034 | 0.093 | 0.72 | 0.185 | | | |
| Apolipoprotein B (SD) | 3 | 0.013 | 0.067 | 0.846 | 0.918 | | | |
| *Disease susceptibility* | | | | | | | | |
| Alzheimer's disease | 18 | 0.035 | 0.013 | 0.009 | 0.783 | – | – | 0.77 |
| Breast cancer | 109 | 0.034 | 0.007 | $7.11 \times 10^{-6}$ | 0.318 | – | – | 0.74 |
| Coronary artery disease | 26 | 0.13 | 0.02 | $3.22 \times 10^{-11}$ | 0.125 | – | – | 2.9 |
| Ischaemic stroke | rs4984814 | 0.012 | 0.003 | $1.39 \times 10^{-5}$ | – | – | – | 0.26 |
| Squamous cell lung cancer | 2 | 0.073 | 0.03 | 0.014 | – | – | – | 1.6 |
| Type 2 diabetes | 22 | 0.036 | 0.015 | 0.02 | 0.247 | – | – | 0.79 |

The 19 traits which were significant in the first step analysis are shown. Exposure, list of exposures tested (for traits in which the betas in the original GWAS were expressed in standard deviations, SD has been added after the name of the exposure). Abbreviations/definitions: SNPs in the IV, the number of variants in the instrumental variable, or the identity of the SNP if < 2. Beta, effects of exposure on lifespan expressed as the log hazard ratio of the Cox model, i.e., parent/offspring effect sizes have been doubled. For traits analysed in SD units, the betas refer to a variation of one standard deviation. CRP, C-reactive protein, DBP, diastolic blood pressure, HDL, high-density lipoprotein, LDL, low-density lipoprotein, SE, the standard error of beta. Egger pleiotropy P refers to the *p*-value from the MR Egger regression. SD, standard deviation of the exposure. Reduced years of life per exposure unit, reduction in lifespan expressed in years per measurement unit of the exposure (not SD units, even for traits where beta is in SD units). A negative number indicates a longer lifespan. Interquartile effect on mortality (years), extrapolated difference in years of life between someone at the 3$^{rd}$ and 1$^{st}$ quartiles of the phenotypic distribution, i.e., a 1.34 SD difference for quantitative traits and 2.2 points on the log(OR) scale for binary traits. SBP, systolic blood pressure

In aggregate, 18 variants in 13 gene regions were identified, of which four were genome-wide significant in the original study, while nine were suggestive (Fig. 3). These lead SNPs were then looked up in our results and compared with the previously reported associations (Supplementary Data 3 and Fig. 3).

Whilst comparable *p*-values were directly apparent, we also wished to compare effect sizes, inter alia to understand whether non-replication in terms of *p*-value arose from lack of power or inconsistency in observed effect. However, this was not straightforward due to the different study designs, principally that we observed hazard ratios for mortality, while other studies observed odds ratios for extreme long-livedness (often for slightly different definitions of cases and controls). We therefore proceeded as follows. The most significant longevity association, *APOE*, was used to estimate the relationship between OR observed in case-control studies and HR observed by us, as follows. For *APOE* variant rs4420638(G) log$_e$ OR for survival beyond age 90 has been estimated elsewhere as −0.33[6] and our observed HR was 0.07, giving an empirical factor of −4.7 to estimate ORs for case-control extreme long-livedness from lifespan HRs. This factor was applied to our observed HR for all the candidate SNPs in Fig. 3, giving an empirical estimate from our data of the OR for extreme long-livedness. Some studies did not report standard errors for their ORs, merely *p*-value and effect estimate. We inferred standard errors, assuming that a two-sided test with a normally distributed estimator had been used.

**Genetic correlations**. We estimated genetic correlations between mortality and other traits using our both parents European ancestry GWAMA summary statistics and the LDHub web portal (http://ldsc.broadinstitute.org/)[49]. As the parent phenotype-offspring genotype GWAS halves the genetic effects[4], both the genetic covariance and sqrt(heritability) estimates are halved, resulting in 1:1 estimation of the offspring-offspring genetic correlation(rg), from parental GWAS-offspring GWAS based estimates of rg. LDHub estimates rg between one test GWAS and

~ 200 traits from metabolomics to common diseases such as cardiovascular disease and lung cancer, using LD score regression[9]. Given their redundancy and number, the metabolomic traits were excluded from the analysis. We added diastolic and systolic blood pressure[50], C-reactive protein (CRP)[12] and breast cancer[11] to the traits present in LDHub[49], using GWAMA summary statistics for these studies provided to us. Each of these was run through the LDHub server in order to estimate the genetic correlations with the other traits while the genetic correlation with lifespan was estimated by using a local run of LD score regression. The Benjamini and Hochberg multiple correction test procedure was applied to determine the statistical significance of the resulting genetic correlations.

We then defined three categories of traits: (a) Meaningfully genetically correlated to mortality if estimated rg > = 0.15 and FDR < 0.05; (b) Not meaningfully genetically correlated to mortality if 95% CI for rg ⊂ [−0.15,0.15]; and (c) Otherwise, insufficient evidence.

After subsetting to only those meaningfully genetically correlated to mortality, we estimated all genetic correlations among those traits; some pairs of traits showed very high correlations. For example, many were genetically correlated to BMI and obesity, we thus used the ICLUST clustering algorithm to cluster the most similar ones. The number of clusters was chosen empirically, by visual inspection. The ICLUST algorithm from the psych R package clusters items hierarchically based on the loading of the items on the factors from factor analysis. Two clusters are then merged together only if by their joining their internal consistency increases. As rotation matrix for the factor analysis we used "promax" which is a high efficiency algorithm which allows correlation between the different factors[51]. Other than to define the initial list, mortality was not included in the clustering analysis. At the same time, some highly correlated traits, which the clustering algorithm sought to combine, appeared to capture distinct clinical aspects and these were therefore kept separate. In particular, we split an education/smoking/

rheumatoid arthritis group into three; we separated CAD out from the cluster for dyslipidaemia and waist-hip ratio (DL/WHR) and we separated breast cancer from age at menarche. The 12 clusters (and their constituent traits) were as follows: obesity (body mass index, body fat, childhood obesity, extreme BMI, obesity class 1, obesity class 2, obesity class 3, overweight, hip circumference, leptin_not adj, leptin_adjbmi, waist circumference & CRP), smoking (cigarettes smoked per day, former vs current smoker and lung cancer), DL/WHR (fasting insulin, extreme waist-to-hip ratio, HDL cholesterol, insulin resistance (from homoeostasis modelling assessment), triglycerides and waist-to-hip ratio), kidney (urinary albumin-to-creatinine ratio and urinary albumin-to-creatinine ratio-non-diabetics), type 2 diabetes (child birth weight, type 2 diabetes, fasting glucose and haemoglobin A1c), blood pressure (systolic and diastolic), happiness (depressive symptoms and subjective well-being), breast cancer, CAD, educational attainment (years of schooling, openness to experience from NEO personality inventory), rheumatoid arthritis and age at menarche.

The resulting correlation matrix amongst clusters was used to estimate partial genetic correlations between the clusters using the matrix inversion method as implemented in the corpcor R package (ISBN: 978-0-470-74366-9).

**Mendelian randomisation**. As a further step to identify which traits affect, rather than merely correlate with, mortality and to determine how much they shorten or lengthen lifespan, we performed a multiple step two-sample Mendelian randomisation (MR) study using summary statistics. We first identified a list of 96 candidate phenotypes selected amongst diseases and disease-associated risk factors (Supplementary Table 2) for which genome-wide association data were publicly available as part of the MRbase package. Data were available for more than one GWAS for a given trait, we selected those which had the largest sample size, were performed on both sexes and were performed in either European or Mixed descent samples. To this list we added other GWAS which were not present in MRbase: diastolic and systolic blood pressure, C-reactive protein and breast cancer. For each of the selected traits, instrumental variables were constructed starting from all SNPs with $p < 5 \times 10^{-8}$. We then performed LD clumping[52] ($r^2 = 0.1$, window = 10 Mb) in order to prune all non-independent SNPS. Some traits had no SNPs below the significance threshold and were thus excluded.

MR was performed using the inverse variance method utilising each of the selected traits as exposures and mortality as outcome. Where the instrument for the trait was composed of a single SNP, we used the Wald ratio instead. We then defined as candidate traits all the phenotypes with a Benjamini and Hochberg FDR < 0.05. We also verified the absence of directional pleiotropy using MR Egger regression, but none of the candidate traits showed statistically significant evidence of pleiotropy once corrected for multiple testing (Supplementary Table 2). Having already corrected for FDR at the previous step, no further adjustment was made for multiple testing in Table 3.

Several traits associated with BMI and obesity were extremely redundant: we thus removed obesity class 1, obesity class 2, obesity class 3, overweight, extreme body mass index, hip circumference, and childhood obesity. Finally, myocardial infarction was removed since 20 of the 22 SNPs composing its instrument were also in the CAD instrument. Supplementary Table 3 summarises the number of SNPs comprising each instrument before and after pruning.

For each significant trait, we estimated the difference in expected years of life between the 25th and 75th phenotypic percentiles. For normally distributed traits this difference corresponds to 1.345 phenotypic standard deviations. For binary traits, the variance of the logistic distribution is constant, this difference instead corresponds to an interquartile distance of 2.2 times the beta coefficient estimated from the logistic regression. Thus, the difference in years between the two considered quantiles was estimated to be:

$$\text{75th} - \text{25th percentile distance} = 1.345 \times \text{SD} \times \beta_{\text{HR}} \times 10$$

For quantitative traits and

$$\text{75th} - \text{25th percentile distance} = 2.2 \times \beta_{\text{HR}} \times 10$$

This measure gives us the difference of expected lifespan between the two risk quartiles expressed in years.

**Data availability**. All relevant data that support the findings of this study are available from the corresponding author upon request or from UK Biobank, LDHub, MRbase, BCAC, CHARGE-CRP[53, 49, 10, 11, 12].

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

## Acknowledgements

We thank the UK Biobank Resource, approved under application 8304; we acknowledge funding from the UK Medical Research Council Human Genetics Unit. We would like to thank the authors of LDHub and MRbase and all the groups who contributed summary statistics there, as well as the BCAC and CHARGE-CRP consortia for sharing their summary statistics with us directly. We also thank Neil Robertson, Wellcome Trust Centre for Human Genetics, Oxford, for use of his author details management software, Authorial, and Tom Haller of the University of Tartu, for tailoring RegScan so we could use it with compressed files. We would also like to thank the researchers, funders and participants of all the contributing cohorts. Study specific acknowledgements are listed in Supplementary Note 1.

## Author contributions

P.K.J., N.Pirastu, K.A.K., K.F., E.H., D.W.C., C.L.K.B., P.R.H.J.T., T.F.H., S.D.G., F.G., T.S.B., A.A., W.Z., C.M.-G., T.M.B., S.T., L.A.L., L.R., A.v.d.S., T.E.G., P.P., L.R.Y., L.F.B., A.P., F.M., M.P.C., G.B., S.M.T., N.A., J.D.F., M.Gögele, J.-J.H., C.M.L., R.M., S.E.M., W.E.O., Y.S., J.A.S., F.J.A.v.R., C.F., N.Pendleton, A.W., N.F., D.C. and T.E. performed the analyses. T.N., S.D.G., W.Z., S.T., L.R.Y., A.P., G.B., I.S., A.D.B., A.C., I.D., J.D.F., I.F., A.G., A.H., J.-J.H., D.M.H., M.A.H., M.A.I., M.J., D.K., L.L., C.M.L., U.L., S.E.M., L.M., R.N., W.E.O., P.A.P., P.P.P., P.M.R., F.R., D.R., R.S., H.S., B.H.S., J.A.S., N.Sotoodehnia, S.H.V., P.V., Y.W., T.W., J.B.W., A.B.Z., T.Lehtimäki, M.K.E., M.P., L.B., N.Pendleton, S.L.K., M.C., D.M.B., B.M.P., C.M.v.D., J.G.W., J.W.J., L.K., A.G.U., N.F., K.E.N., D.R.W., A.M., D.I.B., C.H., D.C., N.G.M., H.C., T.E. and J.F.W. Contributed Data or funding. P.K.J., N.Pirastu, K.A.K., K.F., E.H., K.E.S., D.W.C., T.N., C.L.K.B., P.R.H.J.T., X.S., U.L., K.E.N., N.Sattar, T.E. and J.F.W. designed the analyses. P.K.J., N.Pirastu, K.A.K, P.R.H.J.T., N.Sattar, Z.K. and J.F.W. wrote the manuscript.

## Additional information

**Competing interests:** The authors declare no competing financial interests.

Peter K. Joshi[1], Nicola Pirastu[1], Katherine A. Kentistou[1,2], Krista Fischer[3], Edith Hofer[4,5], Katharina E. Schraut[1,2], David W. Clark[1], Teresa Nutile[6], Catriona L.K. Barnes[1], Paul R.H.J. Timmers[1], Xia Shen[1,7], Ilaria Gandin[8,9], Aaron F. McDaid[10,11], Thomas Folkmann Hansen[12,13], Scott D. Gordon[14], Franco Giulianini[15], Thibaud S. Boutin[16], Abdel Abdellaoui[17], Wei Zhao[18], Carolina Medina-Gomez[19,20], Traci M. Bartz[21], Stella Trompet[22,23], Leslie A. Lange[24], Laura Raffield[25], Ashley van der Spek[20], Tessel E. Galesloot[26], Petroula Proitsi[27], Lisa R. Yanek[28], Lawrence F. Bielak[18], Antony Payton[29], Federico Murgia[30], Maria Pina Concas[31], Ginevra Biino[32], Salman M. Tajuddin[33], Ilkka Seppälä[34], Najaf Amin[20], Eric Boerwinkle[35], Anders D. Børglum[13,36,37], Archie Campbell[38], Ellen W. Demerath[39], Ilja Demuth[40,41,42], Jessica D. Faul[43], Ian Ford[44], Alessandro Gialluisi[45], Martin Gögele[30], MariaElisa Graff[46], Aroon Hingorani[47], Jouke-Jan Hottenga[17], David M. Hougaard[13,48], Mikko A. Hurme[49], M.Arfan Ikram[20],

Marja Jylhä[50], Diana Kuh[27], Lannie Ligthart[17], Christina M. Lill[51], Ulman Lindenberger[52,53], Thomas Lumley[54], Reedik Mägi[3], Pedro Marques-Vidal[55], Sarah E. Medland[14], Lili Milani[3], Reka Nagy[16], William E.R. Ollier[56], Patricia A. Peyser[18], Peter P. Pramstaller[30], Paul M. Ridker[15,57], Fernando Rivadeneira[19,20], Daniela Ruggiero[6], Yasaman Saba[58], Reinhold Schmidt[4], Helena Schmidt[58], P.Eline Slagboom[59], Blair H. Smith[60], Jennifer A. Smith[18,43], Nona Sotoodehnia[61], Elisabeth Steinhagen-Thiessen[40], Frank J.A. van Rooij[20], André L. Verbeek[26], Sita H. Vermeulen[26], Peter Vollenweider[55], Yunpeng Wang[13,62], Thomas Werge[12,13], John B. Whitfield[14], Alan B. Zonderman[33], Terho Lehtimäki[34], Michele K. Evans[33], Mario Pirastu[31], Christian Fuchsberger[30], Lars Bertram[63,64], Neil Pendleton[65], Sharon L.R. Kardia[18], Marina Ciullo[6,45], Diane M. Becker[28], Andrew Wong[27], Bruce M. Psaty[66,67], Cornelia M. van Duijn[20], James G. Wilson[68], J.Wouter Jukema[23], Lambertus Kiemeney[26], André G. Uitterlinden[19,20], Nora Franceschini[46], Kari E. North[46], David R. Weir[43], Andres Metspalu[3], Dorret I. Boomsma[17], Caroline Hayward[16], Daniel Chasman[15,57], Nicholas G. Martin[14], Naveed Sattar[69], Harry Campbell[1], Tõnu Esko[3,70], Zoltán Kutalik[10,11] & James F. Wilson[1,16]

[1]Centre for Global Health Research, Usher Institute for Population Health Sciences and Informatics, University of Edinburgh, Edinburgh EH8 9AG, UK. [2]Centre for Cardiovascular Sciences, Queen's Medical Research Institute, University of Edinburgh, Edinburgh EH16 4TJ, Scotland. [3]Estonian Genome Center, University of Tartu, University of Tartu, Tartu 51010, Estonia. [4]Clinical Division of Neurogeriatrics, Department of Neurology, Medical University of Graz, Graz 8036, Austria. [5]Institute of Medical Informatics, Statistics and Documentation, Medical University of Graz, Graz 8036, Austria. [6]Institute of Genetics and Biophysics "A. Buzzati-Traverso" - CNR, Naples 80131, Italy. [7]Department of Medical Epidemiology and Biostatistics, Karolinska Institutet, SE-171 77 Stockholm, Sweden. [8]Department of Medical Sciences, University of Trieste, Trieste 34100, Italy. [9]Institute for Maternal and Child Health, IRCCS "Burlo Garofolo", Trieste 34137, Italy. [10]Institute of Social and Preventive Medicine, Lausanne University Hospital, Lausanne 1010, Switzerland. [11]Swiss Institute of Bioinformatics, Lausanne 1015, Switzerland. [12]Institute of Biological Psychiatry, Mental Health Centre Sct. Hans, Mental Health Services Copenhagen, Roskilde DK-4000, Denmark. [13]iPSYCH, The Lundbeck Foundation Initiative for Integrative Psychiatric Research, Aarhus DK-8000, Denmark. [14]QIMR Berghofer Institute of Medical Research, Brisbane, QLD 4006, Australia. [15]Division of Preventive Medicine, Brigham and Women's Hospital, Boston, MA 02215, USA. [16]MRC Human Genetics Unit, Institute of Genetics and Molecular Medicine, University of Edinburgh, Western General Hospital, Crewe Road Edinburgh EH4 2XU, UK. [17]Netherlands Twin Register, Department of Biological Psychology, Vrije Universiteit, Amsterdam, Amsterdam Public Health Institute (APH), Amsterdam 1081BT, Netherlands. [18]Department of Epidemiology, School of Public Health, University of Michigan, Ann Arbor, MI 48109, USA. [19]Department of Internal Medicine, Erasmus University Medical Center, Rotterdam 3015 CN, Netherlands. [20]Department of Epidemiology, Erasmus University Medical Center, Rotterdam 3015 CN, Netherlands. [21]Cardiovascular Health Research Unit, Departments of Biostatistics and Medicine, University of Washington, Seattle, WA 98101, USA. [22]Section of Gerontology and Geriatrics, Department of Internal Medicine, Leiden University Medical Center, Leiden 2300RC, The Netherlands. [23]Department of Cardiology, Leiden University Medical Center, Leiden 2300RC, The Netherlands. [24]Department of Medicine, University of Colorado Denver, Anschutz Medical Campus, Aurora, CO 80045, USA. [25]Department of Genetics, University of North Carolina, Chapel Hill, NC 27599, USA. [26]Department for Health Evidence, Radboud Institute for Health Sciences, Radboud university medical center, Nijmegen 6500 HB, The Netherlands. [27]MRC Unit for Lifelong Health & Ageing at UCL, University College London, London WC1B 5JU, UK. [28]Department of Medicine, GeneSTAR Research Program, Johns Hopkins University School of Medicine, Baltimore, MD 21287, USA. [29]Centre for Epidemiology, Division of Population Health, Health Services Research & Primary Care, The University of Manchester, Manchester, Greater, Manchester M13 9PL, UK. [30]Center for Biomedicine, European Academy of Bozen/Bolzano (EURAC), (Affiliated Institute of the University of Lübeck, Lübeck, Germany), Bolzano 39100, Italy. [31]Institute of Genetic and Biomedical Research - Support Unity, National Research Council of Italy, Sassari 07100, Italy. [32]Institute of Molecular Genetics, National Research Council of Italy, Pavia 27100, Italy. [33]Laboratory of Epidemiology and Population Sciences, National Institute on Aging, National Institutes of Health, Baltimore City, MD 21224, USA. [34]Department of Clinical Chemistry, Fimlab Laboratories and Faculty of Medicine and Life Sciences, University of Tampere, Tampere 33014, Finland. [35]Health Science Center at Houston, UTHealth School of Public Health, University of Texas, Houston, TX 77030, USA. [36]Department of Biomedicine–Human Genetics, Aarhus University, DK-8000 Aarhus C, Denmark. [37]Centre for Integrative Sequencing, iSEQ, Aarhus University, DK-8000 Aarhus C, Denmark. [38]Centre for Genomic & Experimental Medicine, Institute of Genetics and Molecular Medicine, University of Edinburgh, Edinburgh EH4 2XU, UK. [39]Division of Epidemiology & Community Health, School of Public Health, University of Minnesota, Minneapolis, MN 55454, USA. [40]Charité Research Group on Geriatrics, Charité, Universitätsmedizin Berlin, Berlin 13347, Germany. [41]Lipid Clinic at the Interdisciplinary Metabolism Center, Charité, Universitätsmedizin Berlin, Berlin 13353, Germany. [42]Institute for Medical and Human Genetics, Charité, Universitätsmedizin Berlin, Berlin 13353, Germany. [43]Survey Research Center, Institute for Social Research, University of Michigan, Ann Arbor, MI 48014, USA. [44]Robertson Center for biostatistics, University of Glasgow, Glasgow G12 8QQ, UK. [45]IRCCS Neuromed, Pozzilli (IS) 86077, Italy. [46]Department of Epidemiology, Gillings School of Global Public Health, University of North Carolina, Chapel Hill, NC 27514, USA. [47]Institute of Cardiovascular Science, University College London, London WC1E 6BT, UK. [48]Center for Neonatal Screening, Department for Congenital Disorders, Statens Serum Institut, Copenhagen 2300, Denmark. [49]Department of Microbiology and Immunology, Fimlab Laboratories and Faculty of Medicine and Life Sciences, University of Tampere, Tampere 33014, Finland. [50]Gerontology Research Center, Tampere, Finland, Faculty of Social Sciences, University of Tampere, Tampere 33104, Finland. [51]Genetic and Molecular Epidemiology Group, Institute of Neurogenetics, University of Lübeck, 23562 Lübeck, Germany. [52]Center for Lifespan Psychology, Max Planck Institute for Human Development, Berlin 14195, Germany. [53]Max Planck UCL Centre for Computational Psychiatry and Ageing Research, Berlin 14195, Germany. [54]Department of Statistics, University of Auckland, Auckland 1010, New Zealand. [55]Department of Medicine, Internal Medicine, Lausanne University Hospital, Lausanne 1011, Switzerland. [56]Division of Population Health, Health Services Research & Primary Care, School of Health Sciences, Manchester Academic Health Science Centre, University of Manchester, Manchester, Greater Manchester M13 9PL, UK. [57]TH Chan School of Public Health, Harvard Medical School, Boston, MA 02115, USA. [58]Austrian Stroke Prevention Study, Institute of Molecular Biology and Biochemistry, Centre for Molecular Medicine, Medical University of Graz, Graz 8010, Austria. [59]Section of Molecular

Epidemiology, Department of medical statistics, Leiden University Medical Center, Leiden 2300RC, The Netherlands. [60]Division of Population Health Sciences, Ninewells Hospital and Medical School, University of Dundee, Dundee DD1 9SY, UK. [61]Cardiovascular Health Research Unit, Division of Cardiology, University of Washington, Seattle, WA 98101, USA. [62]NORMENT, KG Jebsen Centre for Psychosis Research, Institute of Clinical Medicine, University of Oslo, Oslo 0450, Norway. [63]Lübeck Interdisciplinary Platform for Genome Analytics, Institutes of Neurogenetics & Cardiogenetics, University of Lübeck, Lübeck 23562, Germany. [64]Neuroepidemiology and Ageing Research Group, School of Public Health, Imperial College, London W6 8RP, UK. [65]Division of Neuroscience and Experimental Psychology, School of Biological Sciences, Manchester Academic Health Science Centre, University of Manchester, Manchester, Greater Manchester M13 9PL, UK. [66]Cardiovascular Health Research Unit, Departments of Epidemiology, Medicine and Health Services, University of Washington, Seattle, WA 98101, USA. [67]Kaiser Permanente Washington Health Research Institute, Seattle, WA 98101, USA. [68]Department of Physiology and Biophysics, University of Mississippi Medical Center, Jackson, MS 39216, USA. [69]BHF Glasgow Cardiovascular Research Centre, University of Glasgow, Glasgow G12 8TD, UK. [70]Program in Medical and Population Genetics, Broad Institute, Broad Institute, Cambridge, MA 02142, USA. Peter K. Joshi and Nicola Pirastu contributed equally to this work

