## [Peer Review File · Nature Communications]

Reviewer #1 (Remarks to the Author):

Review of Joshi et al.,
Nat. Comm.

This is a nice paper on genetic variants that associate with lifespan. The first part of the paper is an extension of previous work by Joshi et al., 2016 where they searched for SNPs associated with the lifespan of the parents of individuals. In the submitted paper, a total of 606,059 parental lifespans were available for analysis, which is about twice as much as the 2016 paper. Two new, genome-wide significant variants affecting longevity (near 154 HLA-DQA1/DRB1 and LPA) were identified. They are also able to validate five SNPs that had been previously associated with longevity.

Next, the authors perform a genetic comparison of hundreds of diseases and traits to longevity to find causal relationships. They find nine categories of disease/trait that have causal roles in either extending or shortening lifespan.

Comments

The section on Mendelian randomization needs to be revised or deleted. The problem is that Mendelian randomization and LD score regression are conceptually similar. The main difference is that Mendelian randomization compares two phenotypes using just a small number of significant SNPs whereas LD score regression uses a large number of SNPs throughout the genome. LD score regression is generally preferable to MR, so the MR analysis is redundant. I suggest deleting the section using MR analysis. The part about the lifespan effect of the interquartile range is interesting, but this part could be added to the LD scan section in Sup. Table 4.

For the four SNPs associated with parental longevity, did they affect longevity by slowing aging itself or by reducing risk for a disease linked to mortality? For a SNP that slows aging itself, it should show an association across most of the trait clusters that are themselves age-related. An aging SNP might show stronger association to longevity than to any one disease. If the SNP reduces risk for a disease, it should show an association mainly with one disease. A disease SNP should show a stronger association to disease than to longevity.

Minor comments

Confusing, run-on sentence should be revised:

279 using Z scores and equal weights for each study, we found strengthened signals, substantially at APOE

280 (rs4420638, $p = 5.4 \times 10^{-41}$) and slightly in the LPA region (rs1045587, $p = 2.05 \times 10^{-11}$), but no increase

281 in statistical the highest level of statistical significance was observed in the HLA-DQA1/DRB1 region,

282 where there were no SNPs in strong LD with the lead LifeGen SNP, nor was there an increase in

283 significance near CHRNA3/5.

This paragraph needs to be revised:

342 We were also able to show that 9 traits were not meaningfully correlated with lifespan (95% CI for r_g

343 fell wholly within the range [-0.15, 0.15]). These were femoral neck and lumbar spine bone mineral

344 density, serum creatinine, extreme height, height, bipolar disorder, schizophrenia, autism spectrum

345 disorder and platelet count.

Serum creatinine is used to calculate eGFR and is thus nearly synonymous with kidney function, which is one of the main categories linked to longevity. Schizophrenia is well-known to increase mortality. The result that these traits are not linked to lifespan is wrong. This indicates that the rule $[-0.15, 0.15]$ to denote traits that are not meaningfully associated with lifespan is probably wrong.

Reviewer #2 (Remarks to the Author):

Summary

Joshi and colleagues from 70 institutions use population cohorts with genome-wide genotype data to carry out association studies of subject genotype with parental lifespan. They find known longevity associations at APOE and CHRNA3/5, and find previously reported associations at CDKN2A/B, SH2B3 and FOXO3A but at lower statistical significance than in the original reports (and 8 literature SNPs for which associations did not replicate). They also find 2 novel genome wide significant associations with longevity at HLA-DQA1/DRB1 and LPA. The GW significant (APOE and CHRNA3/5) and novel results (HLA-DQA1/DRB1 and LPA) all replicated in the CHARGE-EU 90+ study.

They then use Mendelian randomization to examine genetic correlations between lifespan and other traits. 46 are correlated with mortality, 9 are not, and 55 were inconclusive. They find that the traits of quitting smoking, education, openness to new experience and HDL cholesterol are positively genetically correlated with lifespan, while susceptibility to CAD, amount smoked per day, lung cancer, insulin resistance and body fat are negatively genetically correlated with it. They suggest that the education effect is through quitting smoking, and that obesity acts through CAD. They use the approach of instrumental variables to estimate the per unit effect sizes for BMI units (- 4 mo.) and education years (+11 mo.).

Critique

This paper makes a substantial contribution to our overall understanding of the genetic underpinnings of human lifespan, and will be of high interest to readers within the longevity community and broader readership as well. The overall results are analyzed using very current methods, and are very important (though I would not say they are surprising), and are supported by a huge resource set (lifespans of over 600,000 people) including the UK Biobank and 24 other studies. The Mendelian randomization results are important because, as the authors say, they are carried out "in a framework less hampered by confounding and reverse causality than observational epidemiology". Results of the study have implications for healthcare – for example, higher CRP is genetically correlated with longer life in this study, implying that CRP may rise in response to and be protective against disease, rather than causing it; this brings into question whether development of drugs to reduce it is warranted.

Comments:

Results were meta-analyzed with CHARGE-EU 90+, the studies are given equal weights. Why not weight according to study size?

I suggest inserting in the main manuscript text a copy of the sentence on page 18 (lines 569-70) that the effect sizes were doubled to allow for the association of parental phenotypes in offspring genotypes. This seems too important to only be buried in the methods.

I suggest that the sex and age specific effects of APOE and CHRNA3/5 be mentioned in the main

text.

Minor comments:

The abstract (and page 19) state that 200 traits were considered in the MR analysis, but Supplementary Table 4 and page 11 line 332 refer to 113 traits.

The acronym GWAMA is not defined the first time it is used.

Page 9 line 281 has a grammar error: ...but no increase in statistical the highest level of statistical....

Page 13 smoking quantity definitions are not very clear. They refer to a reduction in life of 4 months for lifelong smoking of 1 cigarette per day. It would seem clearer to state whether this corresponds to 20x4 months (almost 7 years) for lifelong smoking of 1 pack/day.

Page 13 line 400: please be more specific about what you mean by "increased" CRP.

Page 15 line 443 has an erroreffect on in people.....

Page 18 lines 552, 553 seem to state that there were more parents of African ancestry than of European ancestry?

Page 18 line 559 has a spelling mistake.

Page 24 line 846 should be: could use it with compressed files.

Reviewer #3 (Remarks to the Author):

The article presented a large scale Genome-wide association study between genotype and parental mortality. The study was built on an extremely large cohort and well designed, and thus the major conclusions of the article are solid. I have only a few minor suggestions to further improve the article.

1) Since many reported variants from this article are also associated with age-related diseases, it might be helpful to analyze and discuss whether these variants contribute to longevity solely through association with disease risks. In other words, once an individual successfully evades from all fatal incidences, whether or not the genotype still contributes to the healthy lifespan.

2) The large cohort size provided the unprecedented power of this study. However, such study based on mixed populations is susceptible to correlation between mortality and cohort. The authors should carefully discuss such aspect and potential consequential bias.

The Genomic Basis of Human Lifespan

Response to reviewer feedback dated 1 June 2017

We thank all the reviewers for their kind and constructive remarks and set out below our specific responses. We have marked changes in the document. We have made some changes not requested, to meet Nature Communications' submission format: figures and tables have been moved (changes not marked to aid legibility) and the abstract has been condensed.

Reviewer #1 (Remarks to the Author):

The section on Mendelian randomization needs to be revised or deleted. The problem is that Mendelian randomization and LD score regression are conceptually similar. The main difference is that Mendelian randomization compares two phenotypes using just a small number of significant SNPs whereas LD score regression uses a large number of SNPs throughout the genome. LD score regression is generally preferable to MR, so the MR analysis is redundant. I suggest deleting the section using MR analysis. The part about the lifespan effect of the interquartile range is interesting, but this part could be added to the LD scan section in Sup. Table 4.

We agree that MR and LD score regression are similar conceptually: as implicit in your comment they look at the covariance in SNP effects amongst the traits. We note a recent conference abstract jointly from the authors of LDHub and MRBase which says "Both methods can be performed using summary data from publicly available genome-wide association studies, and the two can be combined effectively to screen hundreds of different traits for putative causal relationships."¹ i.e. our strategy. At the same time we accept the distinction between the two appears to be a matter of controversy, George Davey Smith wrote: "Pickrell states, with respect to genetic correlation and MR studies, that "mathematically they are identical". Is this true? In the words of Samuel Beckett, "It is not".² Whilst we accept neither of these citations are peer-reviewed, we think they evidence that there is a school of thought, which not everyone agrees with, that sees the separate value in both approaches. We certainly do not consider ourselves final arbiters in this debate. Practically in terms of actual results, we note observed genetic correlation between CRP and mortality was 0.35, whilst the mendelian randomisation showed an opposite effect size -5.5 months / log mg/l. We continue to believe this is because the pathway from CRP to mortality reduces mortality (the MR result), but common underpinning components of both (in particular susceptibility to heart disease, eg blood lipids) increase mortality and heart disease. This result extends the well known MR result³ that CRP is not implicated in the

¹ *LD hub and MR-base: online platforms for performing LD score regression and Mendelian randomization analysis using GWAS summary data*. Available from: https://www.researchgate.net/publication/315113122_LD_hub_and_MR-base_online_platforms_for_preforming_LD_score_regression_and_Mendelian_randomizati_on_analysis_using_GWAS_summary_data [accessed Jun 2, 2017].

² <http://www.bristol.ac.uk/media-library/sites/integrative-epidemiology/working-papers/UnderstandingMendelianRandomization.pdf>

³ Association between C reactive protein and coronary heart disease: mendelian randomisation analysis based

causation of heart disease, despite its (positive) association with heart disease (and thus further reason to believe the distinction between MR and GC here is real). We also note that reviewer 2 thought the MRs were important (although we recognise they were silent on the question of relative merits of GC vs MR).

We have now acknowledged the overlap of the two methods in the discussion, but also highlight their distinction ~line 465

For the four SNPs associated with parental longevity, did they affect longevity by slowing aging itself or by reducing risk for a disease linked to mortality? For a SNP that slows aging itself, it should show an association across most of the trait clusters that are themselves age-related. An aging SNP might show stronger association to longevity than to any one disease. If the SNP reduces risk for a disease, it should show an association mainly with one disease. A disease SNP should show a stronger association to disease than to longevity.

Agreed. We have used phenoscanner to scan the four lead SNPs for trait associations across wide range of traits and now include a table which shows that for 3, there is a stronger association with a disease than longevity.

Phenoscanner results, discussion and method at ~line 292,411,585

Minor comments

Confusing, run-on sentence should be revised:

279 using Z scores and equal weights for each study, we found strengthened signals, substantially at APOE

280 (rs4420638, $p = 5.4 \times 10^{-41}$) and slightly in the LPA region (rs1045587, $p = 2.05 \times 10^{-11}$), but no increase

281 in statistical the highest level of statistical significance was observed in the HLA-DQA1/DRB1 region,

282 where there were no SNPs in strong LD with the lead LifeGen SNP, nor was there an increase in 283 significance near CHRNA3/5.

Agreed.

Revised at line 243

This paragraph needs to be revised:

342 We were also able to show that 9 traits were not meaningfully correlated with lifespan (95% CI for r_g

343 fell wholly within the range [-0.15, 0.15]). These were femoral neck and lumbar spine bone mineral

344 density, serum creatinine, extreme height, height, bipolar disorder, schizophrenia, autism spectrum

345 disorder and platelet count.

Serum creatinine is used to calculate eGFR and is thus nearly synonymous with kidney function, which is one of the main categories linked to longevity. Schizophrenia is well-known to increase mortality. The result that these traits are not linked to lifespan is wrong. This indicates that the rule [-0.15, 0.15] to denote traits that are not meaningfully associated with lifespan is probably wrong.

Thank you for pointing these out. We now see we have been simplistic in the labelling of the kidney function cluster in “Genetic correlations between trait clusters that associate with mortality”. This is really just albumin to creatinine ratio(ACR) which is the diagnostic indicator of albuminuria. We find that the genetic correlation between Serum creatinine and mortality is low, whereas it is high with albuminuria.

We agree that the lack of measured correlation between schizophrenia and mortality is at first brush surprising. We accept that very early death and schizophrenia are strongly associated, but less so with the sum of components of variation in age at death at older ages, which account for more deaths, (principally the trait clusters in the figure). This point is further emphasized by our study looking at lifespan beyond age 40.

We have revised the label “kidney function” to “albuminuria” Figure 4, ~line 650 We have discussed the diagnostic guidance on albuminuria line 444

We have revised wording in the discussion ~line 444 to highlight schizophrenia might still be leading to very early death but this is not what we are studying line We have also made clear in the introduction ~line 172 that the whole study is considering mortality beyond age 40.

Reviewer #2 (Remarks to the Author):

Comments:

Results were meta-analyzed with CHARGE-EU 90+, the studies are given equal weights. Why not weight according to study size?

Yes, we should have made this clearer. Indeed the omission of a justification was an oversight. The parent imputation method has enabled us to gather a great deal of lifespan information at low cost. However, the CHARGE extreme case-control approach is more powerful for the same n. We compared the power across the two study designs empirically: the Z test statistics for rs4420638 (the most significant SNP overlapping in both studies) were 9.4 and 9.5 for CHARGE and lifeGen respectively, Given this similarity, we felt equal weightings were appropriate.

Wording ~line 894 etc changed to reflect this.

I suggest inserting in the main manuscript text a copy of the sentence on page 18 (lines 569-70) that the effect sizes were doubled to allow for the association of parental phenotypes in offspring genotypes. This seems too important to only be buried in the methods.

Agreed. We have perhaps become too familiar with our own method.

Line 223

I suggest that the sex and age specific effects of APOE and CHRNA3/5 be mentioned in the main text.

Agreed. We have moved this figure to the main text.

Inserted at line 282

Minor comments:

The abstract (and page 19) state that 200 traits were considered in the MR analysis, but Supplementary Table 4 and page 11 line 332 refer to 113 traits.

Sorry - this was a mistake. There are 87 metabolites on MR base that we did not analyse, but were incorrectly counted in the 200.

Reference to 200 removed to help get word count down in abstract

The acronym GWAMA is not defined the first time it is used.

Sorry – now done

line 174

Page 9 line 281 has a grammar error: ...but no increase in statistical the highest level of statistical....

Yes – now corrected

line 354

Page 13 smoking quantity definitions are not very clear. They refer to a reduction in life of 4 months for lifelong smoking of 1 cigarette per day. It would seem clearer to state whether this corresponds to 20x4 months (almost 7 years) for lifelong smoking of 1 pack/day.

Yes agreed

wording changed line 354

Page 13 line 400: please be more specific about what you mean by “increased” CRP.

Yes. Now said “inter-quartile phenotypic range”

line 373

Page 15 line 443 has an erroreffect on in people.....

Yes – now corrected

line 401

Page 18 lines 552, 553 seem to state that there were more parents of African ancestry than of European ancestry?

The counts here are SNPs. African counts are higher due to more SNPs with a Minor allele frequency exceeding 1%, likely due to the greater genetic diversity in Africa ancestry populations.

Page 18 line 559 has a spelling mistake.

yes, sorry

corrected 539

Page 24 line 846 should be: could use it with compressed files.

yes, corrected

line 854

Reviewer #3 (Remarks to the Author):

1) Since many reported variants from this article are also associated with age-related diseases, it might be helpful to analyze and discuss whether these variants contribute to longevity solely through association with disease risks. In other words, once an individual successfully evades from all fatal incidences, whether or not the genotype still contributes to the healthy lifespan.

Yes this is a very interesting point. The phenoscan analysis suggests that the effects of three SNPs are principally mediated through disease (or in the case of APOE a range of traits). We have built on this new analysis in the discussion section to bring out this point.
line 281, 403

2) The large cohort size provided the unprecedented power of this study. However, such study based on mixed populations is susceptible to correlation between mortality and cohort. The authors should carefully discuss such aspect and potential consequential bias.

Agreed. Whilst this aspect should not induce false positives, it does reduce power and estimated effect sizes are influenced.

New wording added to discussion to bring out these points. line 420

Reviewer #1:

Remarks to the Author:

The revised manuscript has responded to my concerns and is suitable for publication.

Reviewer #2:

Remarks to the Author:

I am satisfied with the revisions made.

Reviewer #3:

Remarks to the Author:

The reviewers addressed all my concerns and comments constructively and completely.